# Effects of Intramuscular Injections of Vitamins AD3E and C in Combination on Fertility, Immunity, and Proteomic and Transcriptomic Analyses of Dairy Cows during Early Gestation

**DOI:** 10.3390/biotech11020020

**Published:** 2022-06-09

**Authors:** Wirot Likittrakulwong, Pisit Poolprasert, Worawatt Hanthongkul, Sittiruk Roytrakul

**Affiliations:** 1Animal Science Program, Faculty of Food and Agricultural Technology, Pibulsongkram Rajabhat University, Phitsanulok 65000, Thailand; 2Biology Program, Faculty of Science and Technology, Pibulsongkram Rajabhat University, Phitsanulok 65000, Thailand; poolprasert_p@psru.ac.th; 3Phitsanulok Artificial Insemination and Biotechnology Research Center, Ban krang, Mueang Phitsanulok, Phitsanulok 65000, Thailand; vorawatt@hotmail.com; 4National Center for Engineering and Biotechnology (BIOTEC), National Science and Technology Development Agency, Pathumthani 12100, Thailand; sittiruk@biotec.or.th

**Keywords:** vitamins AD3E, vitamin C, protein expression profiles, gene expression, LC-MS/MS, Ovsynch program, dairy cows

## Abstract

This research aimed to investigate the effects of the intramuscular injection of vitamins AD3E and C in combination immediately before the estrus synchronization program (the Ovsynch program) on conception and pregnancy rates, blood parameters, serum biochemical properties, immune systems, antioxidant parameters, and proteomic and transcriptomic analyses during early gestation in dairy cows. Forty nonlactating multiparous cows were randomly assigned to one of four treatments: (1) C: control with normal saline injection; (2) VAD3E: a single intramuscular injection (I/M) of vitamin AD3E; (3) VAD3EC: injection of both vitamins AD3E and C; (4) VC: a single dose of vitamin C. Blood and serum samples were taken immediately at day 0 (before AI), day 7, and day 14 (after AI for 5 days) from the coccygeal vein. Generally, injections of AD3E and C in combination had no effect on the rate of conception or pregnancy. However, they improved hematological parameters and immune and antioxidant activities. Serum samples were analyzed using LC-MS/MS, and 8190 proteins were identified. Five proteins were successfully validated using the quantitative real-time reverse transcription PCR (qRT-PCR) method. This study found that lymphocyte-specific protein 1 (LSP1, A0A3Q1M894) could be used as a protein biomarker for cows administrated with vitamins AD3E and C.

## 1. Introduction

Free radicals are formed on a continuous basis during metabolism, especially during the ovulatory process. Hence, poor oocyte quality, infertility, and decreasing development of the embryo are caused by free radicals and antioxidant imbalance (oxidative stress) [1,2]. In dairy cows, early and late embryonic losses are the most common in the first and sixth weeks of pregnancy (between days 8 and 21 after fertilization) [3,4]. Vitamins AD3E and C exhibited strong antioxidant and free-radical scavengers and prevented oxidative-stress-induced embryo loss and infertility [5,6]. Vitamin A is essential for vision and bone growth and is necessary for maintaining healthy immune function [7]. Vitamin D, also known as cholecalciferol, plays a crucial role in bone calcification and mineralization by increasing calcium and phosphorus absorption. It aids in the absorption of calcium, iron, magnesium, potassium, and zinc in the intestine. Vitamin D3 may impact the time of first postpartum estrus and the calving interval [6]. Vitamin E is a key player in protecting cellular lipids from oxidation by free radicals, which are potentially harmful consequences of cellular metabolism [8]. In addition to scavenging free radicals, vitamin C also converts vitamin E to its active form [9,10]. Moreover, oxidative stress can be directly scavenged by antioxidant enzymes. These enzymes include superoxide dismutase (SOD), catalase (CAT), glutathione peroxidase (GPx), and glutathione reductase (GR).

In recent years, novel proteomic techniques have greatly aided our understanding of cell activities, physiological and biochemical pathways, and biological mechanisms. Proteomics can determine the gene that encodes the protein and help to speculate on the function of the unknown protein by comparing the amino acid sequence to the genomic database. Furthermore, liquid chromatography combined with tandem mass spectrometry (LC-MS/MS) can be adopted to study very complex protein mixtures for proteomic purposes. Therefore, this research aimed to investigate the effect of intramuscular injection of vitamins AD3E and C in dairy cows just before estrus synchronization program (the Ovsynch program). The cows were then tested for conception and pregnancy rates, blood parameters, serum biochemical properties, immune and antioxidant parameters, and proteomic and transcriptomic analyses during early gestation. The findings provide informative data that could aid in the application of these vitamins in dairy cow reproduction.

## 2. Materials and Methods

### 2.1. Cows and Experimental Design

Experimental procedures and animal care were carried out according to the animal regulations and guidelines of the Department of Livestock Development of Thailand (the Artificial Insemination and Biotechnology Research Center and Use Committee; U1-08313-2562). The percentage rates of Holstein cows in the animals used ranged from 87.50 to 93.75% of all animals in the population, with ages ranging from 48 to 72 months, which were housed in the same commercial dairy operation. Forty nonlactating multiparous cows (body condition score (BCS) = 2.75 ± 0.14 (on a scale of 1 to 5) [11] were randomly assigned to one of four treatments, following a completely randomized design (CRD), as described below:(1)C: the control treatment for which cows were injected with normal saline;(2)VAD3E: cows were administered a single intramuscular injection (I/M) of vitamin AD3E (Phenix, Brussels, Belgium; vitamin A 300,000 IU, vitamin D3 100,000 IU, vitamin E acetate 50 mg; 1 mL per 50 kg live weight use) and progesterone release device (CIDR) was removed at night on day 7 (cows received the CIDR on day 0);(3)VAD3EC: cows received vitamins AD3E and C in combination (a total dose of 1500 mg of vitamin C, ascorbic acid, Q.P., Reasol; 500 mg) at night on day 7;(4)VC: cows were injected with a single dose of vitamin C on the same day as the other treatment.

### 2.2. Estrus Synchronization

Cows were assigned to an estrous synchronization plus timed AI protocol. All animals were cycled and were absent of any clinical disorder. Ovsynch program was conducted with 2 injections of GnRH analogue (250 µg Busereline acetate, Receptal^®^, Intervet, Pune, India) on day 0 and day 9. Cows were synchronized with a CIDR containing 1.9 g of progesterone ((P4); Vetrepharm Canada Inc, Bellevile, ON, Canada), inserted intravaginally for 7 days. On day 7, cows received an injection of PGF2α (25 mg, Dinoprost tromethamine, Dinolytic^®^ Etkin, Istanbul, Turkey) and the CIDR was removed. Cows were artificially inseminated 48 h (day 9) after the injection of PGF2α. The cows were observed for behavioral expression of estrus via visual observation twice a day for 30 min from the administration of the PGF2α injection until the timed AI. Conception and pregnancy rate were determined via palpation per rectum and rectal ultrasound at 45 and 60 days after AI. A brief diagram of the current study is shown in Figure 1.

### 2.3. Blood and Serum Sample Collection

Blood samples were taken immediately from the coccygeal vein on days 0 (before AI), 7, and 14 (after AI for 5 days); deposited into collection tubes with a sterile syringe; and stored at 4 °C until use. After mixing the samples with an anticoagulant solution (ethylene diamine tetraacetic acid (EDTA)), they were used for gene expression analysis. For hematology, hemogram (global count the number of red blood cells (RBC), hemoglobin content, mean corpuscular volume (MCV), and mean corpuscular hemoglobin concentration (MCHC) results were determined as described previously by Oliveira et al. [12]. Blood smears were identified, classified, and read in a 1000× magnification microscope according to their morphological and color characteristics. These included white blood cells (WBC), namely neutrophils (Neu), eosinophils (Eos), lymphocytes (Lin), monocytes (Mon), and platelet cells (PLT). For proteomic analysis, biochemical parameters, hormone levels, and immune activities, blood samples without anticoagulant solution were chosen for proteomic analysis, biochemical parameters, hormone levels, and immune activities. To collect the supernatant, blood samples were allowed to clot at 37 °C for 2 h. Serum samples were centrifuged at 1000× *g* for 15 min at room temperature before being stored at 80 °C [13].

### 2.4. Serum Assay

The biochemical parameters and hormone levels influencing the levels of calcium (Ca), phosphorus (P), blood urea nitrogen (BUN), cholesterol (CHO), glucose (GLU), triglyceride (TG), albumin (ALB), progesterone (P4), estrogen (E), and cortisol hormone in serum samples were determined using an Olympus AU400 chemistry analyzer (Olympus Optical Co., Ltd., Tokyo, Japan) following the manufacturer’s instructions. Immune activities, including alternative complement hemolytic 50 (ACH50) activities, total immunoglobulin (total Ig), and lysozyme activities (LZY), were determined using a method previously described by Incharoen et al. [13] with slight modifications. SOD activities were also determined using a modification of the mentioned method [13], and the enzymatic activity was expressed as the percent inhibition rate. The catalase activity (CAT; EC1.11.1.6) was determined using a catalase assay kit (ab83464; Abcam, Cambridge, UK) according to the manufacturer’s standard procedures. The activity of glutathione peroxidase (GPx; EC.1.11.1.9) was determined using a GPx assay kit (ab102530; Abcam, Cambridge, UK) following the manufacturer’s standard protocols. The activity of glutathione reductase (GR; EC1.8.1.7) was measured using a commercially available assay kit (ab83461; Abcam, Cambridge, UK) according to the manufacturer’s standard processes.

### 2.5. Proteomic Analysis Using LC-MS/MS Technique

The Lowry assay was used to determine the protein concentrations of 120 serum samples using BSA as a standard protein [14]. Disulfide bonds were reduced using 5 mM dithiothreitol (DTT) in 10 mM AMBIC at 60 °C for 1 h, and sulfhydryl groups were alkylated using 15 mM Iodoacetamide (IAA) in 10 mM AMBIC at room temperature for 45 min in the dark. For digestion, samples were incubated at 37 °C overnight with 50 ng/L sequencing-grade trypsin (1:20 ratio) (Promega, Walldorf, Germany). Before injecting the digested samples into the LC-MS/MS, they must be dried and protonated with 0.1 percent formic acid. The data obtained from LC-MS/MS were determined using DeCyder MSTM (Amersham Bioscience AB, Uppsala, Sweden) and MASCOT (MatrixScience; Boston, MA, USA) programs. The Swiss-Prot/TrEMB, Gene Ontology, and signaling pathways were detected using STRAP (Vivek Bhatia, Boston University School of Medicine, Boston, MA, USA) and PANTHER (http://www.pantherdb.org) (accessed on 9 July 2021). The metabolic, regulatory, and biosynthesis aspects of secondary metabolites pathways were analyzed using ipath (http://pathways.embl.ed/iPath2.cgi) (accessed on 9 July 2021) according to the method described elsewhere [13].

### 2.6. Gene Expression Analysis

Total RNA was extracted from blood lymphocytes using a QIAamp RNA blood mini kit (Qiagen, Hilden, Germany). Semi-quantitative RT-PCR was performed as formerly described [15] to measure the levels of interlekin-10 (IL-10), tumor necrosis factor alpha (TNF-α), interferon-stimulated gene 15 (ISG15), superoxide dismutase1 (SOD1), catalase (CAT), and beta-actin mRNA. The gene specific primers are listed in Table 1.

In a MyGo Pro real-time PCR apparatus, the reactions were run in triplicate (IT-IS Life Science Ltd., Mahon, Cork, Ireland). The MyGoPro qPCR program was used to examine the relative expression ratios of mRNAs of these genes (IT-IS Life Science Ltd., Mahon, Cork, Ireland). The 2^−ΔΔct^ calculation was used to assess the real-time PCR findings for gene transcripts [18]. The transcription level of the beta-actin gene was used as a housekeeping gene to adjust the mRNA levels of these genes.

### 2.7. Statistical Analysis

The statistical analysis was carried out using a completely randomized design (CRD). The conception and pregnancy rates were compared between groups via Chi-square analysis. The Chi-square test was statistically performed using the SPSS version 23.0 (SPSS Inc., Chicago, IL, USA). For biological parameters and immune and antioxidant activity levels, the gene expression and relative expression ratio were analyzed using one-way ANOVA. One-way analysis of variance (ANOVA) was also performed using SPSS version 23.0 (SPSS Inc., Chicago, IL, USA). Differences among the groups were analyzed using the new Duncan’s multiple range test (DMRT). Additionally, results are expressed as the mean ± SE.

## 3. Results

### 3.1. Conception Rate and Pregnancy Rate

During the experimental period, the effects of injections of vitamins AD3E and C on the estrus detection percentage as well as conception and pregnancy rates are shown in Table 2. Generally, all values were found to not be significantly different (*p* > 0.05) (Table 2). The rates of conception and pregnancy tended to rise in the VC group (40%, 4 to 10), although cows injected with both vitamins (VAD3EC) tended to show increased conception and pregnancy rates compared to those in the control group.

### 3.2. Hematological Parameters

The hematological parameters, including the WBC, Neu, Lin, and PLT, were influenced (*p* ≤ 0.05) by the injections of vitamins (Table 3). Notably, high WBC and Lin values were observed, and these were statistically different (*p* ≤ 0.01) in all treatments. WBC values in the VAD3EC treatment reached up to approximately 46,580 cells/mm^3^, while WBC values in the VC treatment reached approximately 8460 cells/mm^3^. Additionally, the percentage of Lin in VAD3E and VAD3EC treatments was around 81.40%. Vitamin C affected the PLP values, as evidenced in the VC treatment, with the highest PLP value (376,400 cells/mm^3^). However, the control group (C treatment) exhibited the highest Neu percentage (51.80%), indicating no impact of vitamin administration. In addition, no significant difference was observed for RBC, Ht, Hgb, MCV, MCHC, Eos, and Mon (Table 3).

### 3.3. Biochemical Parameters and Hormone Contents

The injection of vitamins significantly affected (*p* ≤ 0.01) the levels of serum BUN, ALB, and CHO (Table 4). VAD3E exhibited the highest values of BUN and CHO at 28.60 and 168.40 mg/dL (*p ≤* 0.01), respectively. The ALB values differed (*p* ≤ 0.01), with the highest value of 3.32 g/dL found in VAD3EC, while the lowest value of 2.48 g/dL was found in the control (Table 4). However, the levels of serum GLU, TG, Ca, P, P4, E, and cortisol hormone were not affected by the vitamin injections. The cortisol hormone in serum tended to improve when vitamins AD3E and C were administered as compared to the control.

### 3.4. Immune and Antioxidation Activity

For immune activities, ACH50, total Ig, and LZY activities were significantly affected (*p* ≤ 0.01) by the vitamins (Table 5). The highest levels of ACH50, total Ig, and LZY were found in the VAD3E group, amounting to 394.95 U/mL, 2.61 mg/mL, and 499.00 U/mL, respectively, and they were found to be significantly different (*p* ≤ 0.05) (Table 5). In terms of antioxidant activity, the highest levels of serum SOD (94.48%), CAT (0.23 mU/mL), and GR (0.26 U/mL) were found in cows that received vitamins AD3E and C in combination (VAD3EC), while the highest level of serum GPx was observed in the VC group (0.11 mU/mL) (*p* ≤ 0.01) (Table 5). Based on these results, vitamins AD3E and C could improve dairy cow immunity.

### 3.5. Functional Classification of Identified Serum Proteins

To classify the identified serum proteins using Gene Ontology (GO) classification tools, the total percentages of identified serum proteins were divided into groups based on their presence in different biological processes, beginning with the most abundant: cellular process (29.30%), metabolic process (18.40%), biological regulation (17.00%), response to stimulus (8.20%), localization (6.80), signaling (6.10%), multicellular organismal process (4.30%), developmental process (3.90%), immune system process (1.40%), biological adhesion (1.00%), locomotion (1.00%), reproduction (0.70%), reproductive process (0.70%), interspecies interaction between organisms (0.70%), growth (0.20%), biological phase (0.20%), multi-organism process (0.10%), biomineralization (0.10%), and rhythmic process (0.10%) (Figure 2).

### 3.6. LC-MS/MS Identification

Peptide sequences from liquid chromatography–tandem mass spectrometry (LC-MS/MS) were investigated using the *Bos taurus* genome database to determine the protein composition changes in all treatments. There were detectable amounts of 8186, 8075, 8021, and 7928 predicted proteins in the C, VAD3E, VAD3EC, and VC samples, out of a total of 8190 proteins. A total of 7801 proteins were shared in all treatments. One protein was specifically shared in VAD3E and VAD3EC, namely lymphocyte-specific protein 1 (LPS1) (Figure 3).

### 3.7. Differentially Abundant Protein in All Treatments during Induction Times

Abundant protein profile changes from all treatments during the induction time (early, middle, and late at days 0, 7, and 14, respectfully) were analyzed. Lymphocyte-specific protein 1 (LSP1) was specifically expressed in the middle and late (VAD3EC in day 7 and VAD3E in day 14) periods but suppressed in the control and VC treatments during the induction time. Five major proteins were expressed during day 0, day 7, and day 14, including (1) cytokine synthesis inhibitory factor (interlekin-10) (A0A3Q1MVU8); (2) TNF-alpha-induced protein 1 (E1BLB2); (3) ISG15 protein conjugation (F6Q4D3); (4) superoxide dismutase (Mn), mitochondrial (EC 1.15.1.1) (E1BHL1); and (5) catalase (EC 1.11.1.6) (A0A3Q1LKF1) (Table 6). The results showed that the protein marker, LSP, could be a good indicator in cows being induced with vitamins AD3E and C.

### 3.8. Quantitative Real-Time Reverse Transcription PCR (qPCR)

The quantitative real-time reverse transcription PCR (qPCR) technique was used to confirm differentially expressed proteins during induction times. In this experiment, the mRNA expression levels of five differentially expressed genes (IL-10, TNF, ISG15, SOD1, and CAT) were examined over time (days 0, 7, and 14), as shown in Figure 4. The expression levels of IL-10 and ISG15 in each treatment at different interval times revealed that the expression levels of IL-10 and ISG15 in VAD3E, VAD3EC, and VC were clearly decreased during the middle and late intervals (*p* ≤ 0.05) (Figure 4a,c). During the middle interval time, TNFα mRNA was slightly upregulated in cows administrated with vitamin AD3E (*p* ≤ 0.05). Nevertheless, during the late interval times, expression was clearly reduced to a lower level than in the control group (Figure 4b). SOD1 and CAT mRNA levels were significantly upregulated at day 7 (middle interval time) in cows injected with vitamin C according to the expression levels of SOD1 and CAT in each treatment at different times (VC treatment). Nonetheless, the expression was clearly reduced in the late period (Figure 4d,e). Vitamin AD3E also affected the stimulated genes related to immune response, specifically TNF-α in the middle period. Moreover, vitamin C also influenced the stimulated genes related to antioxidant enzymes (SOD1 and CAT) in the middle period.

## 4. Discussion

Oxidative stress has a negative effect on dairy cow fertility, particularly resulting in abnormal follicles, poor oocyte quality, low embryo development, and high embryo mortality [2]. Vitamins AD3E and C are recognized as powerful antioxidants and scavengers of free radicals, and they are essential for the maintenance of good immune functions and normal reproduction in dairy cows. This study found that the percentage of estrus detection, the rate of conception, and the rate of pregnancy were not significant factors in any treatment group, suggesting that the estrous behavior, conception rate, and pregnancy rate were not affected by the administrated vitamins. Likewise, Maldonado et al. [19] reported that vitamins C and E did not significantly increase the pregnancy rate in their cows. Yildiz et al. [20] showed that the administration of a single subcutaneous injection of vitamin E and selenium just before Ovsynch testing had no significant effect on the conception rates of dairy cows. However, the application of those vitamins affected the GPx and SOD activities and progesterone levels. On the other hand, Sarker et al. [21] illustrated that the administration of vitamin AD3E or vitamin minerals induced cyclicity and conception in a higher proportion of anestrus heifers compared to the control group. A study in buffalos found that injections of vitamin AD3E resulted in a 37.5% conception rate compared to 12.5% in the control group [22]. Nutritional deficiency or imbalance was believed to be a primary cause of anestrus, and supplementation of vitamins should be considered for real anestrus [22].

Regarding hematological parameters, this study found that vitamins AD3E and C had an impact on WBC, Neu, Lin, and PLT. Among VAD3EC-treated cows, the highest values were observed for WBC and Lin parameters. The PLP values were considerably impacted by vitamin C, as evidenced by the highest PLP value being observed in VC-treated cows. However, Neu exhibited the highest percentage (51.80%) in the control group. Meanwhile, RBC, Ht, Hgb, MCV, MCHC, Eos, and Mon values did not show any significant changes (refer to Table 3). A similar finding has been documented by other researchers. Mohammed et al. [23] demonstrated the role of vitamin AD3E in regulating incentive hormones during ovulation. The function of estrogen hormone release is to encourage LH, FSH, and prolactin hormones to become more sensitive to strange bodies, resulting in a rise in WBC during the early stages of pregnancy. Oliveira et al. [12] postulated that the total numbers of Neu values were higher in the first and third phases of pregnancy. Higher Lin values were detected in the first and second phases of pregnancy. The Mon values were impacted by the reproductive stages, with greater absolute values in the early phase of pregnancy. For Neu, the changes were most likely caused by physiological events that occurred during the early stages of pregnancy development. This phenomenon promotes an increase in circulating leukocytes, altering the cellular proportions in the differential counts at birth and triggering leukocytosis via neutrophilia in the early stage of pregnancy [24]. Consequently, hematology dynamics can be classified as a biomarker for the different reproductive stages of cows [12].

Serum biochemical and immune parameters are the most important indicators for the health of dairy cows. In this research, high values of serum BUN and CHO were found in cows administered with vitamin AD3E. The percentage of ALB differed with VAD3EC treatment. The rise in the total protein average could be related to the injection of the vitamin mixtures. Vitamin AD3E play an active role in regulating cell osmotic pressure, supporting protein and albumin synthesis, and enhancing its efficiency in cellular protein synthesis [25]. The VAD3E group showed high ACH50, total Ig, and LZY values in this study. The vitamin AD3E mixture might stimulate the immune system by increasing immunoglobulin levels [26].

In terms of antioxidant enzyme activities, the first line of antioxidant enzymatic defense is superoxide dismutase (SOD), which catalyzes the conversion of superoxide radicals to less harmful H_2_O_2_. After this, catalase (CAT) coverts H_2_O_2_ to water. When this mechanism is exhausted, the second line of antioxidant enzymatic defense, primarily GPx, is triggered, which is controlled by selenium availability [27]. In this study, cows that received vitamins AD3E and C in combination showed significantly high levels of serum SOD, CAT, and GR, while the highest level of serum GPx was observed in the VC group. It is concluded that vitamins AD3E and C improve dairy cows’ immune systems and antioxidant enzyme levels. Previous researchers have reported similar findings. Yildiz et al. [20] noted that GPX and SOD activities and plasma progesterone levels were improved by a single subcutaneous injection of vitamin E just before Ovsynch testing in dairy cows. Ali. [28] revealed that lambda-cyhalothrin (LTC), which is a pesticide, administered in two doses for four weeks, significantly reduced the activities of antioxidant defense enzymes (CAT, SOD, GPx, GR, and GST) in a dose-dependent manner when compared to normal rats. Meanwhile, vitamins E, C, and Se were given at the same time as LTC for the same period, and the activity levels of these antioxidant enzymes in brain tissue were significantly higher than in the control group. Apart from injections, there has also been previous research about vitamin supplementation orally. Supplementation with oral vitamins C and E can improve plasma vitamin A, vitamin E, erythrocyte GPx, and glutathione (GSH) levels in basketball players [10]. Vitamin C has been demonstrated to be an important antioxidant for regenerating vitamin E via redox cycling and for increasing intracellular glutathione levels. As a result, vitamin C plays an important role in protecting protein thiol groups from oxidation [29]. Vitamin E (α-tocopherol) protects cells from free radicals, which are potentially harmful byproducts of the body’s metabolism. Vitamins C and E-treated animals demonstrated increases in plasma vitamin A, vitamin C, and vitamin E due to free radical and lipid peroxidation (LP) inhibition [30,31]. As a result, supplementation of vitamins AD3E and C was found to support antioxidant enzyme activity.

This study has reported on the abundance of differential proteins at different interval times in all treatments (refers Table 6). One protein (LSP1, A0A3Q1M894) was present in the VAD3EC group during day 7 (middle induction) and the VAD3E group during day 14 (late induction). Among these proteins, five characterized proteins, namely IL-10, TNF-α, ISG15, SOD1, and CAT, were also detected. The mRNA transcription levels of the identified proteins were verified using qPCR. In this study, the expression levels of IL-10 and ISG15 were significantly reduced in VAD3E, VAD3EC, and VC groups during the middle and late intervals. Regarding TNFα, mRNA levels were marginally upregulated in the middle interval in cows given vitamin AD3E. During the late period, however, expression was obviously lower than in the control group. According to Shirasuna et al. [16], maternal recognition factors such as interferon-tau (TFNT) are identified in cows between days 15 and 19 of pregnancy. TFNT regulated the expression levels of ISG, IL-10, and TNFα. On day 8 of pregnancy after AI, the expression levels of ISGs and IL-10 were upregulated. Meanwhile, TNFα was stable in all groups. As a result, ISGs and IL-10 may be useful target genes for reliable pregnancy indices prior to maternal identification. In this research, the expression levels of SOD1 and CAT mRNA were significantly higher in the VC group on day 7 (middle interval time). Nonetheless, expression was clearly decreased in the late interval time. Enhanced GPX activities may be related to increased gene expression in response to free radical generation, as well as the protective effects of vitamin C and E supplementation on glucose-6-phosphated dehydrogenase activities in erythrocytes [10].

In this study, the transcriptional and translational levels of five expression genes were not correlated. In VAD3E, VAD3EC, and VC groups, the five expression genes showed significantly different mRNA and protein patterns when compared to the control group. The mRNA expression levels of all genes were dramatically changed in the middle interval time (day 7), while proteins level changed in late interval time (day 14). However, the abundance of mRNA and proteins did not correlate well. It was frequently observed that expression occurred at the mRNA level but not at the protein level due to differences in regulation in the transcription and translation steps [32,33]. In addition, Said et al. [34] revealed that mRNA and protein expression levels of SOD and CAT were also downregulated in diabetic animals. The addition of vitamin C, a powerful antioxidant, increased both SOD and CAT activities while having no effect on mRNA or protein expression, implying a role for post-translational modification.

Proteins, protein interactions, and small molecules induced by vitamins A, D, E, and C are critical components in the understanding of molecular and cellular functions. One protein of interest could be identified from the associations among those proteins. In this study, lymphocyte-specific protein 1 (LPS1, A0A3Q1M894) was found to have potential as a protein biomarker. The LPS1 gene encodes a 300 amino acid phosphoprotein that is found in pre-B cells, B cells, concanavalin A (Con A)-stimulated thymocytes, macrophages, and granules [35,36], and appears to be regulated by different combinations of transcription factors in T and B cells. The ability of LSP1 to bind to actin and co-cap with immunoglobulin M (IgM) suggests that this protein may play a role in signal transduction within white blood cells [36].

Five interacting proteins (IL-10, TNF-α, ISG15, SOD1, and CAT), which were predicted by STITCH in this study, were associated with the immune system process, regulation of cytokine production, and antioxidant enzymes (Figure 5). The results in this study confirm that vitamins AD3E and C enhance the immune systems process, regulation of cytokine production, and antioxidant enzymes of dairy cows during early gestation.

## 5. Conclusions

Regarding the injection of vitamins AD3E andC, it had no effect on the rate of conception or rate of pregnancy in dairy cows during early gestation. However, it did improve hematological parameters, biochemistry parameters, and immunological and antioxidant functions. In this study, LSP1 was identified as the protein biomarker for dairy cows injected with vitamins AD3E and C. Using the Ovsynch program, vitamins AD3E and C injection were found to enhance the immune system, cytokine production regulation, and antioxidant enzymes in dairy cows during early gestation.

## Figures and Tables

**Figure 1 biotech-11-00020-f001:**
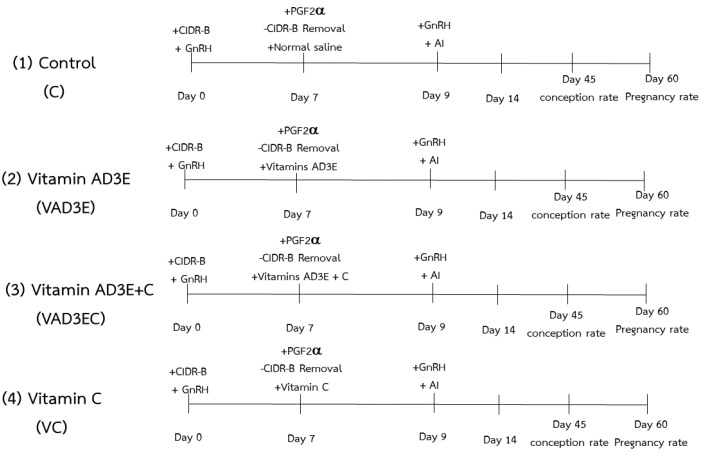
Diagram showing estrous synchronization plus timed AI protocol. (1, C): the control treatment for which cows were injected with normal saline; (2, VAD3E): cows injected with vitamin AD3E; (3, VAD3EC): cows received both vitamins AD3E and C; (4, VC): cows injected with vitamin C; CIDR–B: progesterone release device; GnRH: gonadotropin–releasing hormone analogue; PGF2α: prostaglandin F2 alpha; AI: artificial insemination.

**Figure 2 biotech-11-00020-f002:**
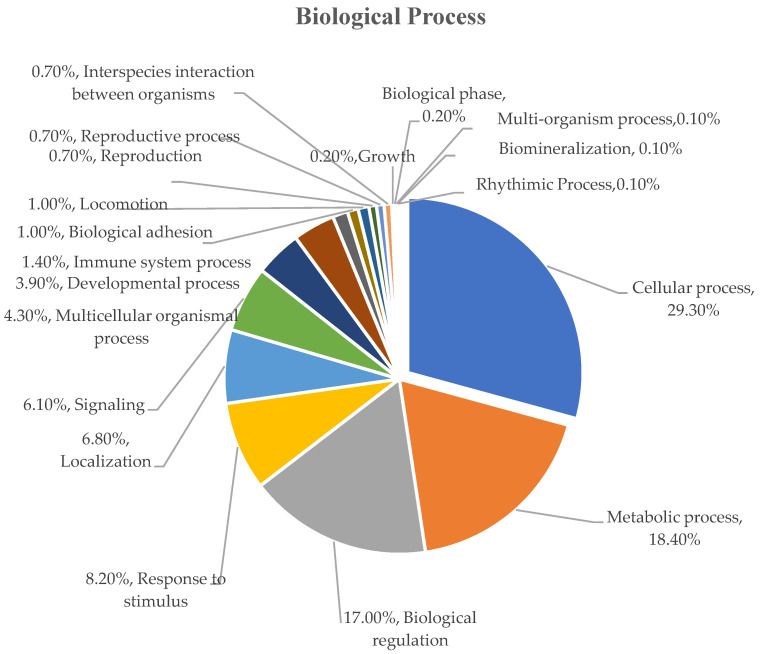
Functional classification of biological processes of serum protein samples from dairy cows during a 14 day period.

**Figure 3 biotech-11-00020-f003:**
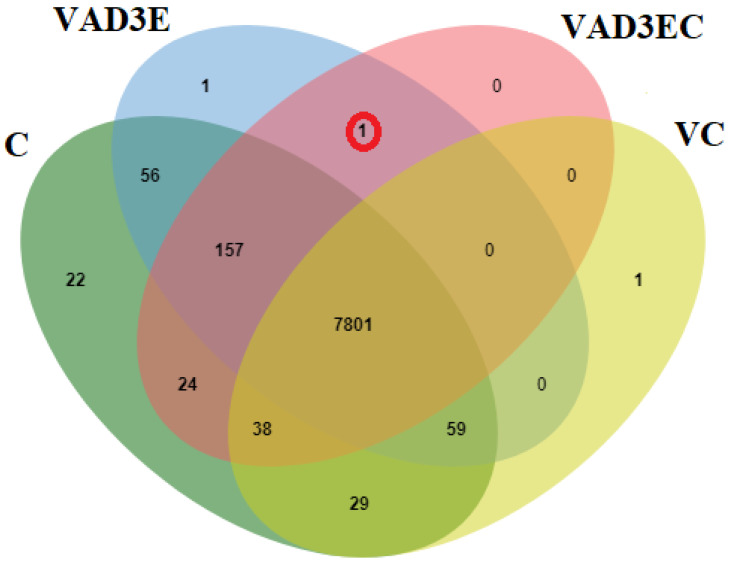
Overlap of identified and abundance proteins in all treatments. C: the control treatment for which cows were injected with normal saline; VAD3E: cows injected with vitamin AD3E; VAD3EC: cows received both vitamins AD3E and C; VC: cows injected with vitamin C; C, green; VAD3E, blue; VAD3EC, pink; VC, yellow.

**Figure 4 biotech-11-00020-f004:**
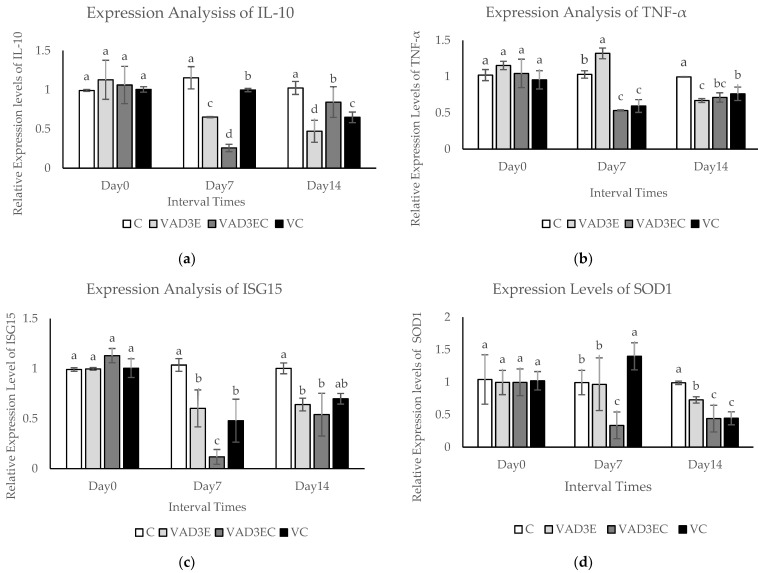
The mRNA expression levels in blood samples of dairy cows over 14 days with different treatments. (**a**–**e**) The mRNA expression levels of IL-10, INF-α, ISG15, SOD1, and CAT in dairy cows with different treatments. Data are presented as means ± SE. Different lowercase letters above each bar indicate significant differences at the same time point (*p* < 0.05). White filled bar: C; light gray filled bar: VAD3E; gray filled bar: VAD3EC; black filled bar: VC. C: the control treatment for which cows were injected with normal saline; VAD3E: cows injected with vitamin AD3E; VAD3EC: cows received both vitamins AD3E and C; VC: cows injected with vitamin C; Day 0: at 0 days (early interval time); Day 7: at 7 days (middle interval time); Day 14: at 14 days (late interval time).

**Figure 5 biotech-11-00020-f005:**
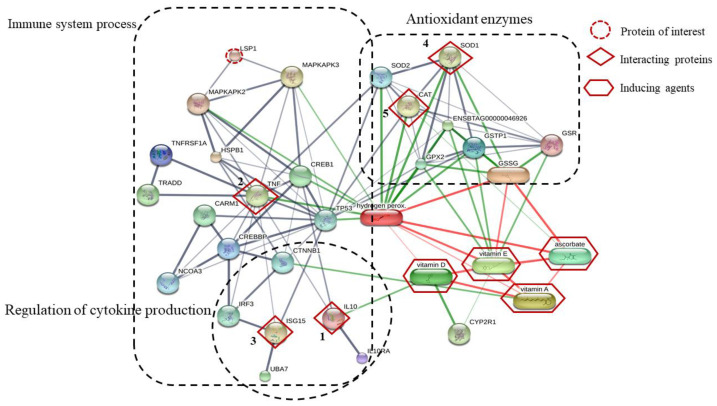
The network of the one identified protein of interest and five interacting proteins predicted by STITCH database based on the following analysis parameters: species (*Bos tarus*); medium confidence score (0.4); active prediction method (no more than 10 interactions). The one protein of interest is lymphocyte-specific protein 1 (LSP1); the five interacting proteins include (1) interleukin 10 (IL-10), (2) tumor necrosis factor-alpha (TNF-α), (3) interferon-stimulated gene 15 (ISG15), (4) superoxide dismutase 1 (Cu-Zn) (SOD1), and (5) catalase (CAT), as well as the inducing agent (vitamin A, vitamin D, vitamin E, and ascorbate). Abbreviation: Glutathione dimer formed by a disulfide bone (GSSG), glutathione reductase (GSR), glutathione S-tranferase P (GSTP1), glutathione peroxidase 2 (GPX2), glutathione peroxidase (ENSBTAG00000046926), superoxide dimutase 2 (Mn) (SOD2), hydrogen peroxide (Hydrogen perox), vitamin D25–hydorxylase (CYP2R1), cellular tumor antigen p53 (TP53), interleukin-10 receptor subunit alpha precursor (IL10RA), ubiquitin-like modifier-activating enzyme 7 (UBA7), catenin beta-1 (CTNNB1), interferon regulatory factor 3 (IRF3), CREB-binding protein (CREBBP), uncharacterized protein (NCOA3), uncharacterized protein (CARM1), tumor necrosis factor receptor type 1-associated DEATH domain protein (TRADD), tumor necrosis factor receptor superfamily (TNFRSF1A), cyclic AMP-responsive element-binding protein 1 (CREB1), heat shock protein beta 1 (HSPB1), MAP kinase-activated protein kinase 3 (MAPKAPK3), MAP kinase-activated protein kinase2 (MAPKAPK2).

**Table 1 biotech-11-00020-t001:** Specific primers used in the experiments.

Gene ^1^	Sequence (5′-3′)	Accession No.	Ref.
IL-10	F: TTCTGCCCTGCGAAAACAA	NM_174088	[16]
	R: TCTCTTGGAGCTCACTGAAGACTCT		
TNFα	F: TGACGGGCTTTACCTCATCT	AF_348421	[16]
	R: TGATGGCAGACAGGATGTTG		
ISG15	F: GGTATCCGAGCTGAAGCAGTT	NM_174366	[16]
	R: ACCTCCCTGCTGTCAAGGT		
SOD1	F: TGTTGCCATCGTGGATATTGTAG	NM_174615	[17]
	R: CCCAAGTCATCTGGTTTTTCATG		
CAT	F: GCTCCAAATTACTACCCCAATAGC	NM_001035386	[17]
	R: GCACTGTTGAAGCGCTGTACA		
Beta-actin	F: GCGTGGCTACAGCTTCACC	AY141970	[17]
	R: TTGATGTCACGGACGATTTC		

^1^ IL-10: interleukin-10; TNFα: tumor necrosis factor alpha; ISG15: interferon-stimulated gene 15; SOD1: superoxide dismutase1; CAT: catalase. Ref: reference.

**Table 2 biotech-11-00020-t002:** Effects of various vitamins on estrus detection, conception rate, and pregnancy rate in dairy cows over 60 days.

Items	C	VAD3E	VAD3EC	VC	*p*-Value
Estrus detection AI%	100.0	90.00	100.00	100.00	0.380
Detections/all cows	10/10	9/10	10/10	10/10	
Conception Rate AI%	10.0	30.0	20.0	40.0	0.446
Detections/all cows	1/10	3/10	2/10	4/10	
Pregnancy Rate%	10.0	30.0	20.0	40.0	0.446
Detections/all cows	1/10	3/10	2/10	4/10	

C: the control treatment for which cows were injected with normal saline; VAD3E: cows were injected with vitamin AD3E; VAD3EC: cows received both vitamins AD3E and C; VC: cows were injected with vitamin C; AI: artificial insemination; conception detected by aggression check via the anus during the 45-day (conception rate) and 60-day (pregnancy rate) periods after artificial insemination.

**Table 3 biotech-11-00020-t003:** Effect of various vitamin on hematology parameters in dairy cows for 14 days.

Items	C	VAD3E	VAD3EC	VC	*p*-Value
RBC (×10^6^ cells/mm^3^)	6.44 ± 0.34	4.96 ± 0.55	5.78 ± 0.49	5.22 ± 0.40	0.138
Ht (%)	30.20 ± 1.53	23.20 ± 2.76	28.40 ± 3.37	24.40 ± 2.99	0.270
Hgb (g/dL)	9.52 ± 0.41	7.46 ± 0.81	8.48 ± 0.88	7.94 ± 0.87	0.299
MCV (fL)	43.38 ± 1.27	47.92 ± 2.85	49.04 ± 1.82	46.82 ± 2.68	0.910
MCHC (g/dL)	31.28 ± 0.33	31.26 ± 0.33	28.74 ± 0.87	32.20 ± 1.48	0.075
WBC (cells/mm^3^)	16,640 ± 7124.79 ^b^	16,620 ± 5956.37 ^b^	46,580 ± 7362.50 ^a^	8460 ± 1148.74 ^b^	0.002
Neu (%)	51.80 ± 12.46 ^a^	24.80 ± 8.35 ^b^	14.80 ± 3.11 ^b^	19.40 ± 4.83 ^b^	0.015
Eos (%)	2.80 ± 0.86	1.40 ± 0.24	1.40 ± 0.40	2.00 ± 0.55	0.280
Lin (%)	43.80 ± 13.07 ^b^	72.40 ± 8.29 ^a^	81.40 ± 1.40 ^a^	77.40 ± 2.23 ^a^	0.015
Mon (%)	1.80 ± 0.20	1.80 ± 0.37	2.20 ± 0.49	1.60 ± 0.40	0.730
PLT (cells/mm^3^)	201,600 ± 29,512.03 ^b^	208,400 ± 8518.22 ^b^	231,200 ± 35,319.12 ^b^	376,400 ± 36,912.87 ^a^	0.002

C: the control treatment for which cows were injected with normal saline; VAD3E: cows injected with vitamin AD3E; VAD3EC: cows received both vitamins AD3E and C; VC: cows were injected with vitamin C; RBC: red blood cells; Hb: hemoglobin; Ht: hematocrit; Hgb: hemoglobin; MCV: mean corpuscular volume; MCHC: mean corpuscular hemoglobin concentration; WBC: white blood cells; Neu: neutrophils; Eos: eosinophils; Lin: lymphocytes; Mon: monocytes; PLT: platelet cells. ^a,b^ Means with different superscripts in a column differ significantly (*p* < 0.05).

**Table 4 biotech-11-00020-t004:** Effects of various vitamins on biochemical parameters of serum in dairy cows over 14 days.

Items	C	VAD3E	VAD3EC	VC	*p*-Value
BUN, mg/dL	4.60 ± 0.51 ^c^	28.60 ± 1.12 ^a^	15.00 ± 1.52 ^b^	8.00 ± 2.07 ^c^	≤0.01
GLU, mg/dL	4.50 ± 0.22	4.60 ± 1.29	2.00 ± 0.32	2.96 ± 0.50	0.057
ALB, g/dL	2.48 ± 0.12 ^b^	3.14 ± 0.18 ^a^	3.32 ± 0.17 ^a^	3.16 ± 0.06 ^a^	0.003
TG, mg/dL	8.20 ± 0.20	9.20 ± 0.37	10.60 ± 1.36	8.40 ± 1.25	0.303
CHO, mg/dL	82.40 ± 2.98 ^c^	168.40 ± 3.97 ^a^	122.60 ± 8.73 ^b^	126.40 ± 7.34 ^b^	≤0.01
Ca, mg/dL	8.36 ± 0.27	8.36 ± 0.42	8.56 ± 0.25	9.24 ± 0.09	0.132
P, mg/dL	9.54 ± 0.74	8.30 ± 0.18	9.40 ± 0.33	8.94 ± 0.88	0.486
P4, ng/mL	0.80 ± 0.25	0.82 ± 0.34	1.02 ± 0.10	1.09 ± 0.39	0.856
E, pg/mL	21.20 ± 10.80	11.60 ± 1.03	40.16 ± 16.83	11.60 ± 1.03	0.168
Cortisol, µg/dL	1.19 ± 0.46	0.43 ± 0.24	0.71 ± 0.55	0.33 ± 0.18	0.190

C: the control treatment for which cows were injected with normal saline; VAD3E: cows were injected with vitamin AD3E; VAD3EC: cows received both vitamins AD3E and C; VC: cows were injected with vitamin C; BUN: blood urea nitrogen; GLU: glucose, ALB: albumin; TG: triglyceride; CHO: cholesterol; Ca: calcium; P: phosphorus; P4: progesterone; E: estrogen. ^a–c^ Means with different superscripts in a column differ significantly (*p* ≤ 0.05).

**Table 5 biotech-11-00020-t005:** Effects of various vitamins on immune and antioxidant activity levels in serum samples from dairy cows over 14 days.

Items	C	VAD3E	VAD3EC	VC	*p*-Value
ACH50, U/mL	294.11 ± 6.24 ^c^	394.95 ± 20.04 ^a^	365.05 ± 22.94 ^ab^	342.14 ± 10.04 ^bc^	0.004
Total Ig, mg/mL	1.34 ± 0.12 ^b^	2.61 ± 0.13 ^a^	2.53 ± 0.20 ^a^	2.59 ± 0.14 ^a^	≤0.01
LZY, U/mL	335.10 ± 13.26 ^b^	499.00 ± 33.60 ^a^	478.83 ± 30.80 ^a^	328.53 ± 24.05 ^b^	<0.01
SOD, inhibition rate, %	85.22 ± 0.73 ^c^	89.26 ± 1.72 ^b^	94.48 ± 0.72 ^a^	91.42 ± 0.49 ^ab^	≤0.01
CAT, mU/mL	0.20 ± 0.00 ^b^	0.01 ± 0.00 ^d^	0.23 ± 0.01 ^a^	0.07 ± 0.01 ^c^	≤0.01
GPx, mU/mL	0.09 ± 0.01 ^c^	0.09 ± 0.00 ^bc^	0.10 ± 0.00 ^ab^	0.11 ± 0.01 ^a^	0.003
GR, U/mL	0.16 ± 0.01 ^c^	0.23 ± 0.02 ^a^	0.26 ± 0.01 ^a^	0.20 ± 0.01 ^b^	≤0.01

C: the control treatment for which cows were not injected with any vitamins; VAD3E: cows were injected with vitamin AD3E; VAD3EC: cows received both vitamins AD3E and C; VC: cows were injected with vitamin C; ACH50: alternative complement hemolytic 50; Total Ig: total immunoglobulin; LZY: lysozyme activities; SOD: superoxide dismutase activities; CAT: catalase activities; GPx: glutathione peroxidase activities; GR: glutathione reductase activities. ^a–d^ Means with different superscripts in a column differ significantly (*p* ≤ 0.05).

**Table 6 biotech-11-00020-t006:** The list of differentially abundant proteins in all treatments during different times.

						Log_2_ Abundance of Protein ^1^											
						Day 0				Day 7				Day 14			
	Uniprot Accession Number	Protein Name	Unique Peptide Sequence	ID Scores	MH+(Da)	C	VAD3E	VAD3EC	VC	C	VAD3E	VAD3EC	VC	C	VAD3E	VAD3EC	VC
1.	A0A3Q1M894	Lymphocyte-specific protein 1	LEQYTQAVEIAGR	7.23	1477.08	0	0	0	0	0	0	17.51	0	0	17.43	0	0
2.	A0A3Q1MVU8	Cytokine synthesis inhibitory factor (interleukin-10)	CHRFLPCENK	9.90	1359.95	15.81	15.43	15.79	14.74	16.20	14.08	14.90	15.30	14.78	17.09	14.97	13.20
3.	E1BLB2	TNF-alpha-induced protein 1	DVIGDEICCWSFYGQGRK	2.61	2190.44	15.67	14.40	16.02	17.99	15.37	14.21	13.39	14.81	14.55	14.76	13.90	14.52
4.	F6Q4D3	ISG15 protein conjugation	KAEVAAEATR	14.75	1045.19	16.60	15.79	16.06	15.29	16.18	15.48	15.16	16.57	16.87	16.58	15.34	15.71
5.	E1BHL1	Superoxide dismutase (Mn), mitochondrial (EC 1.15.1.1)	AGGGAAAVVSLR	5.20	1029.35	16.47	16.50	16.36	16.41	16.59	16.24	16.28	16.30	15.41	16.05	15.54	16.04
6.	A0A3Q1LKF1	Catalase (EC 1.11.1.6)	GIPDGHRHMNGYGSHTFK	12.03	2009.90	14.91	16.18	15.46	16.14	15.120	13.40	15.39	14.87	15.57	20.47	14.83	15.68

^1^ The levels of proteins in each sample are presented as log_2_ values; C: the control treatment for which cows were injected with normal saline; VAD3E: cows were injected with vitamin AD3E; VAD3EC: cows received both vitamins AD3E and C; day 0: early interval time; day 7: middle interval time; day 14: late interval time.

## Data Availability

Not applicable.

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
