# Peer review of "Effects of Intramuscular Injections of Vitamins AD3E and C in Combination on Fertility, Immunity, and Proteomic and Transcriptomic Analyses of Dairy Cows during Early Gestation"

_biotech, 2022, doi:10.3390/biotech11020020_

Round 1

Reviewer 1 Report

Review

for the journal “Biotech (ISSN 2673-6284)

Manuscript ID biotech-1746392

“Effects of Intramuscular Injections of Vitamins AD3E and C 2 Combination on Fertility, Immune, Proteomic and Tran-3 scriptomic of Dairy Cows During Early Gestation”

Authors:   Wirot Likittrakulwong, Pisit Poolprasert, Worawatt Hanthongkul  and Sittiruk Roytrakul

1) Relevant and interesting is the aspiration of the authors to investigate the effects of intramuscular injection of vitamins combination immediately before estrus synchronization program on conception and pregnancy rate, blood parameters, serum biochemical properties, immune systems, antioxidant parameters, proteomic and transcriptomic in dairy cows. The authors found that injections of the combination of AD3E and C had no effect on the rate of conception and pregnancy, but improved hematological parameters, immune and antioxidant activities, and lymphocyte-specific protein 1 (LSP1, A0A3Q1M894) has been proposed for use as a protein biomarker in cows supplemented with vitamins AD3E and C.

-       With this in mind, I suggest that the authors formulate in detail the tasks of the work at the end of the "Introduction" section.

2)  Lines 68-71:  “ Forty nonlactating multiparous cows 68 (body condition score [BCS] = 2.75 ± 0.14 (on scale 1 to 5) [11] were randomly assigned to 69 one of four treatments as described below: (1) C: the control treatment for which cows were not injected with any vitamins”,

-       Does this mean that the treatment was not carried out? This needs to be clarified when describing groups of cows.

3) The same in row 94 and after the other tables.

4) Table 1. Explanation of the abbreviation "Ref." is not given under the table.

5) Lines 165-166: “Chi-square test was statistically performed using the SPSS version23.0 (SPSS Inc., Chicago, IL, USA).”

-       My question is: what statistical programs were used for other statistical tests (ANOVA, Duncan's multiple test)? This should be reported in the statistical analysis section. The application of ANOVA requires certain conditions for data distribution. This should also be described.

6) The work of the authors has practical value. They found that using the Ovsynch program, vitamins AD3E and C injection were found to enhance the immune system, cytokine production regulation, and antioxidant enzymes in dairy cows during early gestation.

The article is interesting, but the adjustments mentioned are recommended.

Sincerely, reviewer.

Author Response

June, 2 2022

Dear Reviewer1

BioTech

            Thank you for your valuable comments. Please find attached the revised version of the manuscript entitled “Effects of intramuscular injection of vitamins AD3E and C combination on fertility, immune, proteomic and transcriptomic of dairy cows early gestation”. We have corrected the manuscript and per your comments and they are marked as red in the revised version. 

Reviewer 2 Report

Dear Authors,

The present manuscript is adequate written and presented. 

Several revision should be applied.  I have highlighted them in the attached file

Author Response

June, 2 2022

Dear Reviewer

BioTech

            Thank you for your valuable comments. Please find attached the revised version of the manuscript entitled “Effects of intramuscular injection of vitamins AD3E and C combination on fertility, immune, proteomic and transcriptomic of dairy cows early gestation”. We have corrected the manuscript as per your comments and they are marked as red in the revised version.

Round 2

Reviewer 2 Report

Dear authors, 

My recommendations were fulfilled.